# Control of scabies in a tribal community using mass screening and treatment with oral ivermectin -A cluster randomized controlled trial in Gadchiroli, India

**Priyamadhaba Behera****¤, Hrishikesh Munshi, Yogeshwar Kalkonde, Mahesh Deshmukh, Abhay Bang***

Society for Education Action and Research in Community Health (SEARCH), Gadchiroli, Maharashtra, India

¤ Current address: Department of Community Medicine and Family Medicine, All India Institute of Medical Sciences, Bhubaneswar, India
* search.gad@gmail.com

## Abstract

### Background

Scabies is often endemic in tribal communities and difficult to control. We assessed the efficacy of a community-based intervention using mass screening and treatment with oral ivermectin in controlling scabies.

### Methods/ Findings

In this cluster randomised controlled trial, 12 villages were randomly selected from a cluster of 42 tribal villages in Gadchiroli district. In these villages, trained community health workers (CHWs) conducted mass screening for scabies. The diagnosis was confirmed by a physician. Six villages each were randomly allocated to the intervention and usual care arm (control arm). In the intervention arm (population 1184) CHWs provided directly observed oral ivermectin to scabies cases and their household contacts. In the usual care arm (population 1567) scabies cases were referred to the nearest clinic for topical treatment as per the standard practice. The primary outcome was prevalence of scabies two months after the treatment. Secondary outcomes were prevalence of scabies after twelve months of treatment and prevalence of impetigo after two and twelve months of treatment. Outcomes were measured by the team in a similar way as the baseline. The trial was registered with the clinical trial registry of India, number CTRI/2017/01/007704.

In the baseline, 2 months and 12 months assessments 92.4%, 96% and 94% of the eligible individuals were screened in intervention villages and 91.4%, 91.3% and 95% in the usual care villages. The prevalence of scabies in the intervention and usual care arm was 8.4% vs 8.1% at the baseline, 2.8% vs 8.8% at two months [adjusted relative risk (ARR) 0.21, 95% CI 0.11–0.38] and 7.3% vs 14.1% (ARR 0.49, 95% CI 0.25–0.98) at twelve months The prevalence of impetigo in the intervention and usual care arm was 1.7% vs 0.6% at baseline, 0.6% vs 1% at two months (ARR 0.55, 95% CI 0.22–1.37) and 0.3% vs

**Data Availability Statement:** All relevant data are within the manuscript and its Supporting Information files.

**Funding:** This project was funded from the grant to SEARCH by the Sir Ratan Tata Trust and Navajbai Ratan Tata Trust (IN) (grant number NRTT 20140610; www.tatatrusts.org/topics/navajbai-ratan-tata-trust), the Grant was awarded to A.B., Director, SEARCH. The salary of H.M. was supported by Sir Ratan Tata Trust (grant number 20160818). The funders had no role in study design, data collection and analysis, decision to publish, or preparation of the manuscript.

**Competing interests:** The authors have declared that no competing interests exist.

0.7% at 12 months (ARR 0.42, 95% CI 0.06–2.74). Adverse effects due to ivermectin occurred in 12.1% of patients and were mild.

## Conclusions

Mass screening and treatment in the community with oral ivermectin delivered by the CHWs is superior to mass screening followed by usual care involving referral to clinic for topical treatment in controlling scabies in this tribal community in Gadchiroli.

## Author summary

Scabies is a skin infestation caused by a mite. It leads to disabling itching and sometimes serious bacterial infections. Scabies is endemic in tribal communities. Skin creams and lotions are available to treat scabies but patients may not apply them thoroughly. We conducted a controlled trial in 12 villages to test if an oral medicine called Ivermectin can reduce scabies in a tribal community in central India. In six randomly selected villages (control arm), trained community health workers (CHWs) screened for scabies, physician confirmed the diagnosis and referred patients to the nearest clinic for treatment with skin creams. In the remaining six villages (intervention arm) the patients of scabies and their contacts were provided supervised treatment with oral ivermectin by the CHWs after the diagnosis. Number of scabies cases were evaluated at two and twelve months after these treatments. The risk of scabies was reduced by 79% and 51% at the end of two and twelve months in villages where oral Ivermectin was used compared to villages where patients were referred to receive skin creams. Adverse reactions due to Ivermectin were mild. Screening of individuals and treating scabies with oral Ivermectin by CHWs can be a useful method to reduce scabies in tribal communities.

## Introduction

Scabies is a contagious disease caused by a mite *Sarcoptes scabiei* which mostly spreads through direct, skin-to-skin contact. It is associated with significant morbidity due to itching, sleep disturbance, reduced ability to concentrate and risk of serious secondary bacterial infections due to *Streptococcus pyogenes or Staphylococcus aureus* [1]. It also leads to social stigmatization and healthcare expenditure [2]. Scabies is a major global public health problem and affects close to 130 million people worldwide at any time [3]. Global age-standardised DALYs per 100 000 people from scabies was 71·11 (95% CI 39·77–116·03) for both sexes [4] and the indirect effects of glomerulonephritis and rheumatic heart disease due to secondary bacterial infections add to this burden [2,5]. Endemic scabies is a disease of tropical developing countries having poor health care facilities. The burden of scabies is particularly high in various tribal communities in India ranging from 6.9–20.6% [6–7]. In 2017, the World Health Organisation (WHO) added scabies as Neglected Tropical Diseases to its portfolio. The recommendation was made to respond to the high burden of scabies and its complications, particularly in areas with limited access to health care, and in the light of new public health control strategies for reducing the burden [8].

 Standard management of scabies involves applying topical agents such as permethrin, malathion, benzyl benzoate or sulphur ointment all over the body. However, discomfort due to topical treatment, inadequate application to local lesions, poor adherence to treatment among

contacts of scabies cases are major barriers for effective use of topical therapy [9]. Oral therapy can offset some of the challenges associated with topical therapy and can potentially increase compliance resulting in better control of scabies in communities. There is increasing interest in oral ivermectin in the treatment of scabies. The safety of ivermectin has been well studied in large scale disease eradication programmes for other parasitic diseases. For example, in the past 21 years, more than 1 billion doses of ivermectin tablets have been distributed for both onchocerciasis and filariasis, at doses of 100–200 μg/kg safely [10]. Recent studies have shown the beneficial effect of mass drug administration of ivermectin on scabies in Fiji [11]. In this cluster randomised controlled trial we tested a hypothesis that after mass screening for scabies, a community-based intervention which used directly observed therapy (DOT) with oral ivermectin would be superior to screening followed by referal to existing healthcare facilities in reducing the prevalence of scabies in a tribal community in central India where scabies is endemic.

## Methods

### Ethics statement

The study protocol was approved by the Institutional Ethical Committee of SEARCH constituted as per the guidelines of the Indian Council of Medical Research. Written informed consent was obtained from the *Gram Sabha* (local village council) of the study villages before enrollment. In addition, written informed consent was obtained from adult participants and parents for minors (age <18 years) before administering ivermectin to the entire family of cases with scabies. This consent procedure was approved by the ethical committee. The trial was registered with the Clinical Trial Registry of India (CTRI/2017/01/007704) [12].

### Study setting

The study was conducted in a tribal block (Dhanora) of Gadchiroli, one of the most underdeveloped districts of India. Close to 75% of its land is covered by forests [13]. Gadchiroli has a population of 10,72,942 [13]. Forty percent of the population is tribal, belonging to the scheduled tribes as described in the Constitution of India. In Gadchiroli district the major tribes are Gond, Madiya, Pardhan, Kolam and Urao. These people remain geographically, socioeconomically and politically marginalized and their health indicators remain poorer compared to nontribal people [14]. Society for Education, Action and Research in Community Health (SEARCH) is a not-for-profit organization working in tribal and rural regions of Gadchiroli since 1986. It has a field practice area of 48 tribal villages where a majority of the population belongs to the Gond tribe. The average distances of these villages is 38 km from Gadchiroli town which is the district headquarter. As per the census conducted by SEARCH in 2013 the population of these villages was 12,188. In these 48 villages SEARCH operates a mobile medical unit (MMU) aided by the National Rural Health Mission, Government of India which visits a village every month and provides treatment free of cost to villagers.

The study was a two arm cluster randomized controlled trial conducted from January 2017 to January 2018. The primary outcome was the reduction in the prevalence of scabies among the people of intervention villages as compared to usual care villages two months after the intervention. The secondary outcomes were reduction in the prevalence of scabies 12 months after the intervention and reduction in the prevalence of impetigo two and twelve months after the intervention. Physician assessors who diagnosed scabies and impetigo were blinded to the intervention. Sample size was calculated using a method described by Hayes and Bennett for cluster-randomised trials [15]. We assumed a between-cluster correlation coefficient of 0.2 (k = 0.2) as there were no data available from prior studies. We anticipated that the prevalence

of scabies in this area will be 7%. This was based on the minimal prevalence reported from studies in tribal communities in India and our own experience during providing primary clinical care in this area [6–7]. Considering, a loss to follow up of 20%, the study needed 6 villages per arm (average population of 250 per village) to detect 50% reduction in the prevalence of scabies with 80% power and 95% confidence ($\alpha = 5\%$). Since we used mass screening followed by treatment of cases and contacts (MSAT) instead of mass drug administration (MDA), we chose 50% reduction as plausible for estimation of sample size. Higher reduction ranging from 60–90% has been observed with MDA [11].

Out of 48 villages in the field practice area of SEARCH, six villages had population less than 100 and were excluded. 12 villages were selected randomly from the remaining 42 villages. The size of villages included in random selection ranged from 100 to 737 with an average population of 254. Prevalence of scabies was estimated in 12 villages in the baseline survey. Using computer generated randomization, the statistician of SEARCH assigned these 12 villages to the intervention (n = 6) or the usual care arm (n = 6).

## Baseline measurement

Village wise list of residents in the 12 study villages was obtained from the population register of SEARCH which is updated annually. All residents from these villages were eligible to participate. All individuals who provided consent to participate in the study were included. For minors (<18 years) consent was sought from parents. The exclusion criteria at individual level included- ivermectin use at the time of evaluation or in the past seven days, use of topical 5% permethrin in the past seven days, seriously ill patients or those with known allergy to any of the components of the allocated drug regimen. A team of community health workers (CHWs) of SEARCH who were resident of the villages, field supervisors and physicians were trained to obtain informed consent, screen individuals and diagnose scabies. In the intervention arm, the field supervisors were also trained to follow up those treated with ivermectin or permethrin and record adverse drug reactions. The baseline survey to assess the prevalence of scabies was conducted in two phases. In the first phase, a house to house survey was carried out by trained CHWs in the 12 villages in the study between January and February 2017 using a pre-tested questionnaire. If the residents were not available during the first round of screening, the CHW made two more visits to their houses to contact them and labeled them as absent if they could not be contacted after three visits. Screening questionnaire inquired about current itching, presence of papules or pustules and the location of lesions. Those with either itching or presence of papules or pustules at sites typical for scabies were considered screen positive. In the second phase, the screen positive individuals were evaluated by a trained physician for the diagnosis of scabies by making home visits within 7 days after the visit by the CHW. Among the participants who were classified as screen negative by CHWs, about 5% were selected by systematic random sampling, and evaluated by a trained physician for scabies. All the 153 participants classified as screen negative by CHWs at baseline were found to be free of scabies by the physician. For the residents not available during first round evaluation, the physician made two more visits to their houses in next seven days before labeling them as absent. Residents who could not be contacted after three visits of the CHWs or physicians or did not provide consent to participate in the study were excluded from the study.

Scabies was defined as the presence of more than one pruritic, inflammatory papule with a typical anatomical distribution involving web spaces of fingers, hands, wrists, elbows, knees, trunk, groin or ankles [16–17]. The severity of scabies was determined by the number of lesions identified as mild ($\leq 10$), moderate (11 to 49) and severe ($\geq 50$) [10]. Infected scabies was diagnosed if there were pus filled sores or crusted sores within the lesions of scabies [11].

Any suspicious crusted scabies cases were referred to the rural hospital of SEARCH for evaluation. The discomfort due to scabies was evaluated using a structured questionnaire by the physician. Impetigo was defined as the presence of a papular, pustular, or ulcerative lesion surrounded by erythema [11].

### Intervention arm

Mass screening and treatment for scabies was implemented in this arm as follows. After initial screening by the CHWs and diagnosis of scabies cases by a physician, the CHWs visited all scabies cases in their homes weighed them and administered ivermectin at a dose of 200 μg/kg given in the form of 3mg tablets as a directly observed treatment. For scabies cases this dose was repeated after 14 days. A single dose ivermectin (200 μg/kg) was also administered to all household members unless contraindicated. Outside the trial intervention, ivermectin is sparsely available in this area. Contraindications to the use of ivermectin included weight less than 15 kg, pregnant and breastfeeding women [18], individuals with neurological conditions-such as Parkinson's disease or cerebral palsy and individuals taking anticonvulsant therapy. Ivermectin was administered under direct observation by the CHWs. Ivermectin was replaced with 5% topical permethrin lotion in those with contraindication to administration of ivermectin. Two whole body (except face) applications of topical permethrin were advised with a gap of 14 days. Those with infected scabies and impetigo were treated with topical application of 0.5% gentian violet paint by the CHWs in addition to treatment with ivermectin [19]. All the treated individuals (cases and family members) were asked to report any adverse drug reaction to the CHW from the time of administering the first dose till 14 days after the second dose. Information on adverse drug reaction was obtained by direct questioning from those receiving the treatment in the intervention arm using a pre-tested questionnaire. This was done by a field supervisor between 7–14 days after ivermectin or permethrin treatment and was confirmed by a physician. Adverse events were classified as serious if they were life threatening, led to hospitalization, or resulted in persistent or substantial disability or death.

### Usual care arm

In this arm, the cases diagnosed with scabies and their contacts were referred to receive usual care for scabies currently available in these villages. The usual care in these villages included treatment by a MMU which visited these villages once in a month or referral to the nearest government-run primary health centre (PHCs) for scabies management according to the standard guideline in these centres. The mobile medical unit provided 5% permethrin for treatment of scabies to patients and household contacts while the treatment available in the PHCs included 25% benzyl benzoate. The information on treatment of scabies of patients referred was extracted from the monthly clinic register of the MMU. The availed treatment history was also obtained from the participants in the follow-up visits during the trial period.

Scabies cases with severe co-morbidities (fever above 101°F, patients showing systemic signs and symptoms of infection, those with severe protein energy malnutrition, children with infected scabies or impetigo, patients with more than 40 lesions) were referred to the rural hospital of SEARCH for further management in both the arms.

### Follow-up

A mass screening and diagnosis similar to the baseline evaluation was carried out by the trained CHWs and physician in the intervention and usual care arm between March and April 2017, two months after the intervention. Another survey was conducted in January 2018-twelve months after the intervention. In the follow-up at two months, scabies cases in the

intervention area received ivermectin or permethrin (in those with contraindication for ivermectin) while those in the usual care area received 5% topical permethrin provided by the MMU of SEARCH. In the twelve month follow up visit, scabies cases in both intervention and usual care areas received oral ivermectin and where contraindicated, topical permethrin.

### Statistical analysis

Information on the village residents was obtained from the population census conducted by SEARCH. The results were reported as proportions with 95% confidence intervals (95% CI) for prevalence of scabies. Risk of scabies in the intervention arm at two and twelve months compared to that in the usual care arm was calculated after adjusting for age, sex and presence of scabies at baseline. Among the participants who could be evaluated in all three evalutions- at baseline, at two months and at twelve months, the disappearance rate of scabies (the proportion of individuals with scabies at baseline who did not have scabies at two months follow-up survey or twelve months follow-up survey) and the appearance rate (the proportion of individuals without scabies at baseline who manifested scabies at two months follow-up survey or twelve months follow-up survey) were also calculated. For multivariate analyses we used random effects logistic regression models to account for clustering. We assessed the effect of the intervention on the prevalence of scabies after adjusting for age, sex and prevalence of scabies at baseline and that on the rate of disappearance and appearance of scabies after adjusting for age and sex. All the statistical analyses were intention-to-treat and and were conducted using Stata version 12.0 (Statacorp, Texas, USA).

### Results

A total of 2751 individuals from 12 study villages (1184-intervention arm, 1567-usual care arm) were eligible to participate in the study. Out of these, 1094 (92.4%) and 1432 (91.4%) from intervention and usual care arms were screened at baseline, 1137(96.0%) and 1430 (91.3%) were screened at two months follow-up survey and 1113 (94.0%) and 1501 (95.8%) were screened at twelve months follow-up survey respectively. Some individuals who were absent at baseline evaluation got enrolled at the survey conducted at two months (intervention-135, usual care-161) and twelve months (intervention-113, usual care-174). Also, some individuals who were present at baseline evaluation were not available at two months (intervention-92, usual care-163) and at twelve months (intervention-177, usual care-205). Out of 92 participants, who were lost to follow-up at two months, 40 participants were tracked again at the twelve months follow-up survey in the intervention arm. Similarly, out of 163 participants, who were lost to follow-up at two months, 102 participants were tracked again at the twelve months follow-up survey in the usual care arm (Fig 1).

The age, sex and family size were comparable in the intervention and the usual care arm (Table 1). The population of villages in the usual care arm was higher than that of the intervention arm. The baseline prevalence of scabies was comparable in the intervention arm (8.4%, 95% CI, 7.1% - 10.5%) and the usual care arm (8.1%, 95% CI, 6.7% - 9.5%) (Table 2). In the intervention arm and the usual care arm, the distribution of mild (42.4% vs. 43.1%), moderate (34.8% vs. 35.3%), and severe (22.8% vs. 21.6%) scabies was also comparable (S1 Table). There were no cases of crusted scabies in intervention and the usual care area.

In the intervention arm, 88 out of 92 scabies cases and 208 family contacts received first dose of treatment under direct observation from the CHWs. Four scabies cases did not receive ivermectin. A women opted to apply leaves of local *Neem* tree to her two children and two cases, after the confirmed diagnosis of scabies, travelled to other area for work before the CHW reached their home to provide the treatment. Ivermectin was administered to 241

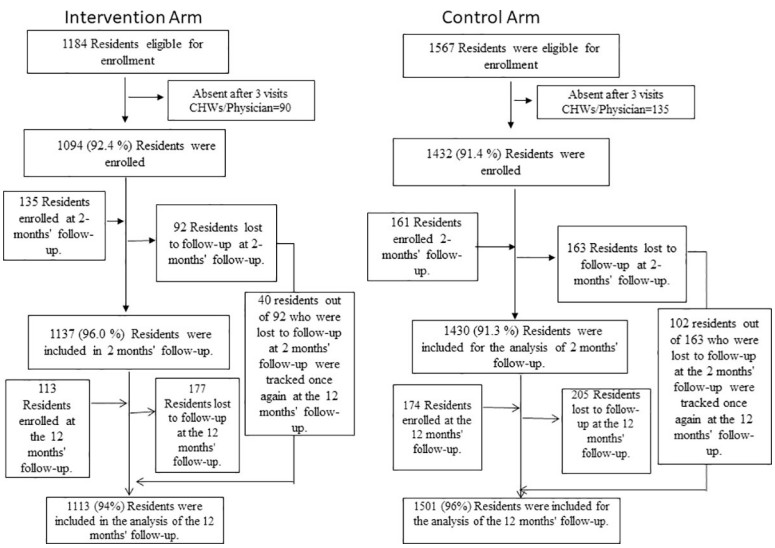

**Fig 1. Enrollment and Follow-up.**

(81.4%) individuals while 55 (18.6%) received 5% permethrin due to contraindications to ivermectin administration. All (n = 88) scabies cases identified at baseline evaluation received the second dose of treatment (ivermectin or permethrin if ivermectin was contraindicated) after 14 days under direct observation of CHWs. In the control arm forty percent (47/116 participants) of referred scabies cases availed treatment in the MMU or a PHC where scabies cases as well as their contacts received treatment.

In the follow up at two months, the prevalence of scabies in the intervention and usual care arm was 2.8% (95% CI 1.9–3.8) and 8.8% (95% CI 7.4–10.3). Surprisingly, at twelve months

**Table 1. Baseline characteristics of the participants.**

| Variables | Intervention arm | Control arm |
|---|---|---|
| **No. of Villages** | 6 | 6 |
| **No. of Households** | 225 | 320 |
| **Median family size (range)** | 4 (1–17) | 3 (1–16) |
| **Population** | 1184 | 1567 |
| **% Female** | 50.8 | 52.5 |
| **Age** | No. (%) | |
| < 5 yrs | 125 (11.4) | 181 (12.6) |
| 5–14 yrs | 140 (12.8) | 145 (10.1) |
| 15–29 yrs | 342 (31.3) | 443 (31.0) |
| 30–44 yrs | 254 (23.2) | 338 (23.6) |
| 45–59 yrs | 144 (13.2) | 209 (14.6) |
| > 60 yrs | 89 (8.1) | 116 (8.1) |
| **Total population included** | 1094 (100) | 1432 (100) |
| **Coverage (%)** | 92.4% | 91.4% |
| **Weight (Kg)- Mean (SD)** | 38.6 (14.8) | 37.7 (14.7) |

All diagnosed cases of scabies had itching. The median duration of itching was 30 days ranging from 2 to 365 days. More than half (54.8%) of all diagnosed cases of scabies had sleep deprivation. The median duration of sleep deprivation was 15 days ranging from 2 to 120 days (S2 Table).

**Table 2. Prevalence of scabies at the baseline, at two and twelve month follow-up evaluation in the two arms of the study.**

| Arm | Baseline evaluation | | Two month Follow-up | | ARR at two month follow-up[#] | Twelve month Follow-up | | ARR at twelve month follow-up[#] |
|---|---|---|---|---|---|---|---|---|
| | No. of patients/ total population | % prevalence (95% CI) | No. of patients / total population | % prevalence (95% CI) | ARR (95% CI) | No. of patients / total population | % prevalence (95% CI) | ARR (95% CI) |
| Intervention | 92/1094 | 8.4 (7.1–10.5) | 32 / 1137 | 2.8 (1.9–3.8) | **0.21 (0.11–0.38)**[**] | 81/1113 | 7.3 (5.9–9.0) | **0.49 (0.25–0.98)**[*] |
| Control | 116 / 1432 | 8.1 (6.7–9.5) | 127/ 1430 | 8.8 (7.4–10.3) | | 212/1501 | 14.1 (12.5–16.0) | |

#adjusted for age, sex and scabies at baseline

**p-value<0.0001

*p-value = 0.047

the prevalence increased in both the arms—7.3% (95% CI 5.9–9.0) in the intervention and 14.1% (95% CI 12.5–16.0) in the usual care arm but remained significantly lower in the intervention arm compared to the usual care arm. The adjusted relative risk (ARR) of scabies was 0.21 (95% CI 0.11–0.38) at two months and 0.49 (95% CI 0.25–0.98) at twelve months in the intervention arm compared to the usual care arm (Table 2).

The rate of disappearance of scabies in the intervention arm vs usual care arm was 91.9% vs 65.3% at two months and 88.3% vs 78.7% at twelve months. The ARR for the disappearance of scabies was 6.02 (95% CI 2.15–16.86) and 2.23 (95% CI 0.89–5.56) at two and twelve months in the intervention arm (Table 3). The appearance rate of scabies was significantly higher in the usual care arm compared to intervention arm both at two (6.0% vs. 1.9%) and twelve months (13.0% vs. 6.3%). The estimated ARR for appearance rate of scabies in the intervention arm was 0.27 (95% CI 0.15–0.48) and 0.48 (95% CI 0.24–0.96) at two and twelve respectively (Table 3).

The prevalence of impetigo in the intervention and usual care arm was comparable at baseline (1.7%vs. 0.6%), at two (0.6% vs.1.0%) and at twelve months follow-up (0.3%, vs. 0.7%). The ARR of impetigo was 0.55 (95% CI 0.22–1.37) at two months and 0.42 (95% CI 0.06–2.74) at twelve months follow up in the intervention arm compared to the usual care arm (Table 4).

Adverse drug reactions (ADRs) were mild and comparable among the individuals who received oral ivermectin or permethrin (12.1% vs. 12.7%) in the intervention arm (S3 Table).

**Table 3. Disappearance and appearance of scabies in the intervention and control arms at 2 and 12 month follow-up.**

| Follow-up at months | Arm | Disappearance of Scabies | | | Appearance of scabies | | |
|---|---|---|---|---|---|---|---|
| | | Scabies present at baseline evaluation | Scabies absent | ARR for disappearance of scabies[#] | Scabies absent at baseline | Scabies present | ARR for appearance of scabies[#] |
| | | (n) | (n) | ARR (95% CI) | (n) | (n) | ARR(95% CI) |
| 2 months | Intervention | 85 | 78 | **6.02 (2.15–16.86)** | 916 | 17 | **0.27**[**] **(0.15–0.48)** |
| | Control | 96 | 62 | | 1174 | 70 | |
| 12 months | Intervention | 77 | 68 | 2.23 (0.89–5.56) | 841 | 53 | **0.48**[*] **(0.24–0.96)** |
| | Control | 94 | 74 | | 1120 | 145 | |

#adjusted for age and sex.

**p-value<0.0001. Only those who were present in both the baseline and two months follow-up as well as at baseline and twelve months follow-up are included in this analysis.

*p-value = 0.040

**Table 4. Prevalence of impetigo at baseline, two and twelve months follow-up evaluation in the two arms of the study.**

| Arm | Baseline evaluation | | Two month Follow-up | | ARR at two month follow-up[#] | Twelve month Follow-up | | ARR at twelve month follow-up[#] |
|---|---|---|---|---|---|---|---|---|
| | No. of patients/ total population | % prevalence (95% CI) | No. of patients / total population | % prevalence (95% CI) | ARR (95% CI) | No. of patients / total population | % prevalence (95% CI) | ARR (95% CI) |
| Intervention | 18/1094 | 1.7 (1.0–2.6) | 7/ 1137 | 0.6 (0.3–1.3) | 0.55 (0.22–1.37) | 3/1113 | 0.3 (0.1–0.8) | 0.42 (0.06–2.74) |
| Control | 9 / 1432 | 0.6 (0.3–1.2) | 15/ 1430 | 1.0 (0.6–1.7) | | 11/1501 | 0.7 (0.4–1.3) | |

#adjusted for age, sex and impetigo at baseline

Itching was the most common ADR affecting 4.5% and 11.0% individuals administered ivermectin and permethrin respectively. Dizziness (3.3%), nausea (2.5%), headache (0.8%) and burning sensation over skin (0.8%) were the other ADRs reported by cases receiving ivermectin. One individual had rashes (1.8%) after application of permethrin. There was no serious ADR in the intervention arm.

## Discussion

To our knowledge this is the first cluster randomised controlled trial using mass screening and treatment (MSAT) approach to control scabies in an endemic tribal area. In our study, the adjusted relative risk of scabies was reduced by 79% at two months and 51% at the end of twelve months in the intervention arm. The adjusted relative risk of appearance of scabies was also lower by 73% at the end of two months and 52% at the end of twelve months in the intervention arm compared to the usual care arm. Mass screening followed by directly observed treatment with oral ivermectin delivered by community health workers was superior to screening followed by referral to existing healthcare facilities in reducing the prevalence of scabies at the community level in this tribal community in Gadchiroli.

Oral ivermectin is emerging as an important agent to control scabies at the community level. In terms of efficacy one study has shown that oral ivermectin and topical permethrin (5%) were equally efficacious [20] while another showed that oral ivermectin to be less efficacious than topical compounds such as permethrin or benzyl benzoate [21]. Also, a recently published Cochorane review by Rosumeck et al estimated that ivermectin was associated with slightly lower rates of complete clearance after one week compared with permethrin 5%, but after 2 weeks, there was no difference in efficacy [22]. Oral ivermectin may improve control of scabies at the community level by improving compliance [11]. In the SHIFT trial conducted in Fiji, [11] mass drug administration of ivermectin was superior to that of permethrin with 94% reduction in the prevalence in ivermectin group compared to 62% reduction in the permethrin group. The reduction in the prevalence of scabies at the end of 12 months (51%) was lower in our study than that reported in the SHIFT trial (94%).

The additional reduction observed in the SHIFT trial could be from mass drug treatment where those with subclinical scabies would have received treatment leading to reduced transmission and prevalence of scabies at the community level. In contrast, in our study, only cases of scabies and their contacts were treated and those with subclinical scabies might have been missed. In addition, the prevalence of scabies in our study was lower at baseline (~8%) compared to that in the SHIFT trial where it was about 35%. This might have contributed to a lower reduction in the prevalence in our study.

At a population level different strategies can be used to control scabies where it is endemic. Mass drug administration (MDA) and mass screening followed by treatment of cases and contacts (MSAT) are two important strategies which are commonly used. MDA has been shown to reduce the prevalence of scabies in Fiji [11]. However, this strategy may have certain limitations such as higher costs and challenges to ensuring complete coverage in a community where there is migration and in non-island population [8]. Also, it is unclear at what prevalence thresholds mass drug administration for scabies is appropriate [23]. In our study, we considered both MDA and MSAT as options. As there are no guidelines regarding using MDA or MSAT for a given prevalence of scabies, based on our experience of working closely with this community, local logistics and our clinical experience in treating scabies we apriorily decided to consider MDA if the prevalence of scabies is more than 10% in the community and MSAT if the prevalence is <10% [12]. As the baseline prevalence of scabies was <10% in both the arms, we used MSAT of cases and contacts.

In our study, the prevalence of scabies did not reduce in the usual care arm at two months, and the prevalence of scabies was higher at twelve months in both the arms compared to that at two months. The lack of reduction in the prevalence in the usual care arm indicated a failure of referred patients to go to clinic for further treatment. Fifty-nine percent of the referred scabies patients from the usual care arm did not avail the treatment. The potential reason for this failure could include lack of prioritization by those with scabies to seek care or not being available in the village when MMU visited or challenges to travel to PHCs for further treatment. The increased prevalence in both the arms at 12 months is likely to be due to highly conducive environment for transmission of scabies (e.g. higher rainfall and humidity, cold weather) and migration in this community [24–25]. Our findings are similar to that reported by Kearns TM et al where the authors reported difficulty in sustaining reduction in the prevalence of scabies in an Australian aboriginal communities [26]. In the community where we conducted this study, a large number of children from the villages included in the trial study in tribal residential schools where scabies is quite common. When these children visit their families in the villages they often become a source of infection. Similarly family members visiting from other villages could be a potential source of scabies. On the contrary in the SHIFT trial, a sustained reduction in the prevalence of scabies was achieved. The possible reasons for the sustained reduction at 12 months in this trial could be limited mobility due to island setting and use of MDA instead of MSAT.

Although ivermectin helped to reduce the prevalence of scabies in our study, factors related to development such as housing, hygiene, sanitation, and education need to be addressed to ensure that scabies can controlled on a sustained basis. However, until such broader developmental issues can be addressed mass screening and treatment seems to be an effective strategy to control scabies in under developed tribal communities to improve quality of life as well as health.

A reduction in the prevalence of impetigo was observed in the SHIFT trial after MDA of oral ivermectin. Even though there was a reduction in the prevalence of impetigo in our study in the intervention arm both at two months and twelve months follow-up, this was not statistically significant. In our study the prevalence of impetigo is much lower than that reported in the SHIFT trial where it was 24.6% at baseline and 8% after treatment with oral ivermectin [11]. One of the reasons for a lower prevalence in our study could be the availability of health care provided through a mobile medical unit operated by SEARCH which visits the villages in both the intervention and usual care arm every one month and provides treatment free of cost. Other possible explanation for the findings could be a lower risk of secondary bacterial infection among scabies patients in the community where this study was conducted. A lower prevalence of impetigo was also observed in a school-based study among tribal children in Tamil

Nadu in India, where 16.9% of children had scabies, but only 2.3% had impetigo [27]. It is also possible that the study community had a lesser risk of having impetigo in general than the population in Fiji, where the SHIFT trial was conducted. A systematic review of the global childhood population prevalence of impetigo found that the prevalence was lower among school children in Asia (7.3%) against Oceania (29.7%), where Fiji is located [28].

The study has several strengths. It is the first cluster randomised trial which provides the evidence for scabies control with oral ivermectin using a community-based MSAT approach in a tribal community. The study was conducted in a well defined population in a demographic surveillance site. Coverage of treatment and follow-up rates were also high possibly due to involvement of community health workers.

There are also certain limitations. The potential limitation of our trial could be that the findings may be generalisable only to the tribal underserved areas with poor access to health care. The diagnosis of scabies in our study was made using clinical examination and standardized definitions and we did not use dermoscopy which could be a limitation. However, a systematic review of diagnostic methods of scabies has shown low accuracy of dermoscopy in diagnosing scabies and clinical diagnosis remains an accepted practice in field studies [29]. Also, dermoscopy for the diagnosis of scabies has high specificity but low sensitivity [30]. Compared to MDA a two step MSAT strategy where a physician was involved in confirming the diagnosis is more resource intensive. However, in communities where the prevalence of scabies is moderate (5–10%), MDA may not be justifiable. Given the safety of ivermectin, training CHWs to screen, diagnose and treat scabies remains a possibility. Under the National Rural Health Mission, trained community-level workers called Accredited Social Health Activists (ASHAs) are available at the village level [31]. Further studies would be needed to evaluate if CHWs can be trained to diagnose and treat scabies and cost effectiveness of MDA versus MSAT in controlling scabies in low resource setting where prevalence of scabies is <10%.

Our study shows that community-based strategy of mass screening followed by directly observed treatment of scabies provided by a community health worker is an effective strategy to control scabies in a tribal community. Identifying optimal frequency of screening and treatment of scabies with oral ivermectin, prevalence thresholds to determine whether MDA or MSAT will be suitable for a given community and cost effectiveness of these approaches emerge as further research questions.

## Supporting information

**S1 CONSORT Checklist. CONSORT Checklist.**
(DOC)

**S1 Table. Distribution of severity of scabies (based on number of skin lesions) at baseline and end line evaluation.**
(DOCX)

**S2 Table. Symptoms reported by scabies patients in 12 villages at baseline evaluation (n = 208).**
(DOC)

**S3 Table. Adverse drug reactions due to ivermectin and permethrin in the intervention Arm.**
(DOCX)

**S1 Data. Scabies trial wide long data and data info.**
(XLSX)

## Acknowledgments

We thank Drs Suraj Mhaske, Shraddha Dhakulkar, Manveen Kaur, Rashmi Kulkarni, Jalindar Baravkar as well as Mahadeo Satpute, Haridas Sakhare, Jitendra Shahare, Swati Meshram, Shubhangi Meshram, Purushottam Jengthe and Vinod Alam for assistance in the field. We are thankful to the village councils and people in 12 villages for their participation in the study.

## Author Contributions

**Conceptualization:** Priyamadhaba Behera, Abhay Bang.

**Data curation:** Yogeshwar Kalkonde, Mahesh Deshmukh.

**Formal analysis:** Priyamadhaba Behera, Yogeshwar Kalkonde, Mahesh Deshmukh.

**Funding acquisition:** Abhay Bang.

**Investigation:** Priyamadhaba Behera, Hrishikesh Munshi.

**Methodology:** Priyamadhaba Behera, Yogeshwar Kalkonde, Abhay Bang.

**Project administration:** Priyamadhaba Behera, Hrishikesh Munshi, Yogeshwar Kalkonde.

**Resources:** Priyamadhaba Behera, Hrishikesh Munshi, Abhay Bang.

**Software:** Mahesh Deshmukh.

**Supervision:** Yogeshwar Kalkonde, Mahesh Deshmukh, Abhay Bang.

**Visualization:** Priyamadhaba Behera.

**Writing – original draft:** Priyamadhaba Behera.

**Writing – review & editing:** Priyamadhaba Behera, Hrishikesh Munshi, Yogeshwar Kalkonde, Mahesh Deshmukh, Abhay Bang.

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
