## [Decision Letter · Decision Letter 0]

26 Sep 2020

Dear Dr. Bang,

Thank you very much for submitting your manuscript "Control of scabies in a tribal community using mass screening and treatment with oral ivermectin -A cluster randomized controlled trial in Gadchiroli, India" for consideration at PLOS Neglected Tropical Diseases. As with all papers reviewed by the journal, your manuscript was reviewed by members of the editorial board and by several independent reviewers. In light of the reviews (below this email), we would like to invite the resubmission of a significantly-revised version that takes into account the reviewers' comments. 

This paper reports the results of a well conducted cluster randomized clinical trial for the control of scabies comparing directly observed treatment with ivermectin against usual care. The fact that it was directly observed treatment should be included in the title and mentioned in the abstract. The authors should go through the journal's data availability policy (https://journals.plos.org/plosntds/s/data-availability) and comply with it.

We cannot make any decision about publication until we have seen the revised manuscript and your response to the reviewers' comments. Your revised manuscript is also likely to be sent to reviewers for further evaluation.

Sincerely,

Arunasalam Pathmeswaran

Associate Editor

Dennis Bente

Deputy Editor

This paper reports the results of a well conducted cluster randomized clinical trial for the control of scabies comparing directly observed treatment with ivermectin against usual care. The fact that it was directly observed treatment should be included in the title and mentioned in the abstract. The authors should go through the journal's data availability policy (https://journals.plos.org/plosntds/s/data-availability) and comply with it.

Reviewer's Responses to Questions

**Key Review Criteria Required for Acceptance?**

**Methods**

-Are the objectives of the study clearly articulated with a clear testable hypothesis stated?

-Is the study design appropriate to address the stated objectives?

-Is the population clearly described and appropriate for the hypothesis being tested?

-Is the sample size sufficient to ensure adequate power to address the hypothesis being tested?

-Were correct statistical analysis used to support conclusions?

-Are there concerns about ethical or regulatory requirements being met?

Reviewer #1: No concerns regarding these points in the instructions.

Reviewer #2: The objectives are clearly stated, and the selected study design does justice to meeting the objectives. The calculation of sample size needs further elaboration - an issue which has been highlighted in the feed back provided to the authors. The ethical and regulatory requirements were met with satisfaction.

Reviewer #3: -Are the objectives of the study clearly articulated with a clear testable hypothesis stated? Yes

-Is the study design appropriate to address the stated objectives? Yes

-Is the population clearly described and appropriate for the hypothesis being tested? Yes

-Is the sample size sufficient to ensure adequate power to address the hypothesis being tested? Yes

-Were correct statistical analysis used to support conclusions? Yes

-Are there concerns about ethical or regulatory requirements being met? No

- It is not clear what the definition of tribal is in the context of this study. Please describe what makes this population unique.

- Line 116 states that there are 48 tribal villages where the organization works and these are the villages that were included in the random selection for the study. It is not clear however, what proportion of the total tribal population these 48 villages make up in Gadchiroli. Please include, the total population of Gadchiroli, total tribal population and total population within the 48 villages.

- Line 118, please give a range and average of village sizes included in the random selection.

- Line 123, please justify why scabies prevalence at 2 months was selected as the primary outcome. Itch can persist up to 3 months post successful treatment (Romani et al., 2015, NEJM) and has been used in this study as part of the diagnostic criteria. Residual symptoms may persist to produce false positives at 3 months.

- Delete “in the intervention arm” in line 126.

- Line 130, 6 clusters per arm is very few for this study design. Please justify your reasoning for selecting this sample size.

- MSAT stands for mass screening and treatment, suggest using this correct expansion but describing what it means for this particular study.

- Please elaborate if these 48 villages that were included for random selection were comparable, was there a variation in population size per cluster?

- Line 133, what about in the control arm? Usual care with resource supplementation from a study seems to lead to reductions in scabies prevalence.

- Line 144, what is the definition for minors in this study?

- Can you explain why there were these exclusion criteria for screening in lines 145-147? As this is a community health centred project, it is important to consider all members of the community as everyone is a potential source of transmission.

- Can the authors please add information about is ivermectin available in this area?

- Line 149, can the authors describe in more detail about the criteria by which cases of scabies were diagnosed?

- Line 152, it would be helpful to have a flowchart that outlines the two-phase process and the interval of time between them, which is unclear.

- Line 154, describe how valid is the two-phase approach, this is a critical issue

- Lines 162-164 belongs in results

- How did you classify people that were residents of the village? How long would they have to live there? How did you record visitors?

- Was permethrin applied as directly observed therapy for those that were ineligible for ivermectin?

- Please explain why only a single dose of ivermectin was provided to the household members of a scabies case?

- Lines 190-191, was gentian violet paint the only treatment provided for those with impetigo? Is this standard of care in this region? Was a referral to medical facilities offered to receive antibiotics?

- Under the “Usual care arm” section, please specify if usual care included treatment of contacts

- Line 196, when were cases referred in the usual care arm? Was this during the screening visit?

- -Line 202 “patientsbeing toxic” is unclear, suggests changing to “patients showing systemic signs and symptoms of infection” or be clear about the signs and symptoms that met criteria for referral to the rural hospital. 

- Lines 207-213, the study provided treatment of scabies cases found at 2 months. This is a critical issue as this changes the intervention to baseline and 2 months rather than a single intervention for measuring the 12 month outcome.

- Line 223-225, more detail is required for the calculation of cluster-adjustment in the analysis

- How was consent obtained for adults that lacked capacity to provide informed consent?

**Results**

-Does the analysis presented match the analysis plan?

-Are the results clearly and completely presented?

-Are the figures (Tables, Images) of sufficient quality for clarity?

Reviewer #1: (No Response)

Reviewer #2: The analysis matched the presented analysis plan. However there was a bit of a confusion with one of the presented findings - it has been highlighted in the review.

Reviewer #3: -Does the analysis presented match the analysis plan? Yes

-Are the results clearly and completely presented? Yes

-Are the figures (Tables, Images) of sufficient quality for clarity? Yes

- Line 248- There seems to be a big difference in the sample size of the intervention and control group by over 30%, did this have an effect on the results? This issue should be included in the discussion.

- Please add 95% confidence intervals to your prevalence figures- was the increase in 12-month prevalence from baseline in the control group significant?

- It would be good to describe the age-specific prevalence of scabies and impetigo that you found as these findings might shed some light on the resurgence of scabies at 12 months.

- Line 307-309, is the comparison appropriate?

**Conclusions**

-Are the conclusions supported by the data presented?

-Are the limitations of analysis clearly described?

-Do the authors discuss how these data can be helpful to advance our understanding of the topic under study?

-Is public health relevance addressed?

Reviewer #1: (No Response)

Reviewer #2: (No Response)

Reviewer #3: -Are the conclusions supported by the data presented? Yes

-Are the limitations of analysis clearly described? Yes

-Do the authors discuss how these data can be helpful to advance our understanding of the topic under study? Yes

-Is public health relevance addressed? Yes

- Line 371-372, this is not the main reason why this study is important, it is important because it is the first trial using MSAT - please re-write this sentence to reflect that

- Lines 377-380, it is valuable that the end prevalence was lower in the intervention arm compared to control, but ultimately, the prevalence at the end was the same as baseline in both groups, please explain why you think that the intervention was successful when there was no improvement in prevalence by 12 months.

- Line 391-392, please specify, at which time point prevalence fell by 55%. 

- Line 393-395, yes that is right. Perhaps this paper speaks to the importance of subclinical cases as later spreaders. However, please discuss how a lower baseline prevalence might have been a factor as well.

- Such large fluctuations of scabies prevalence after the intervention in both the intervention and control (standard care group) make the results difficult to interpret. Besides migration and not treating subclinical infestation, is there any data that might demonstrate seasonal fluctuations?

- Line 399, this is a simplification. There are more strategies that the two that are stated here

- Line 402-403, also island population versus non-island population

- Line 404-405, this is true, please also refer to WHO informal consultation on a framework for scabies control meeting report published in July 2020

- Line 408 “apriorily”- what does this mean?

- Lines 413-415, please rephrase, this sentence is very confusing

- Line 415-417, it is a strength of this study that this data was collected however it is missing in the results and methods section, please include it there. How long was presentation to health facilities following the visit monitored for? How was it monitored?

- Lines 420-421, please elaborate what environmental factors in the population make it highly conducive to scabies transmission? 

- Line 435-436, “ In a study in Australia, 95% of scabies cases were attributed to environmental factors”, delete this

- Is there any reason that explains higher scabies prevalence at 12 months than baseline in the control group? Surely the community would be more aware of the condition with the study visits.

- Line 465-466, this is a critical point and needs to be expanded.

- Line 467, please define what “moderate’ scabies prevalence is here.

- Line 473, was data on acceptability collected? The authors cannot claim this if no acceptability data was collected

- Lines 474-477, this is an excellent way to end

**Editorial and Data Presentation Modifications?**

Reviewer #1: Authors should use spell and grammar checks.

Reviewer #2: The language, though appropriate, needs fine tuning. There are grammatical errors and continuity errors in the manuscript.

Reviewer #3: A flowchart figure of the screening and diagnosis process would be helpful, as mentioned above.

**Summary and General Comments**

Reviewer #1: PNTD-D-20-01303

The primary objective of this study was to compare the impact of selective treatment of persons with scabies and their household contacts with ivermectin vs standard care (referral and topical treatment) on scabies prevalence 2 months after intervention. They found that ivermectin was superior to standard care for this and for several secondary outcomes. While innovation and originality are not strengths of this study, the results are clear and the report is well written. However, I believe that the study suffers from serious design flaws. These and other issues are listed below: 

1. The study compared two very uneven interventions (selective treatment with ivermectin for cases and household contacts vs. usual standard of care (topical therapy), which for 60% of cases and contacts was no care whatsoever. This should not have been a surprise to the investigators who knew that residents of the study area have limited access to medical care (PHC or MMU). Is it scientifically valid to compare a DOT intervention with an intervention that required active participation by participants? Consider this analogy: Two interventions were offered to drowning people. One group were provided with floatation devices, while the other group was instructed to swim 500 meters to an assistance center where flotation devices would be provided on request. Which intervention worked better? 

2. More than 10% of participants were less than 5 years of age, while ivermectin is only approved for ages 5 and above (or 15 kg). 

3. Methods should include a definition for "impetigo". 

4. It seems that outcomes of the two interventions were compared based on ITT and not corrected for crossovers due to pregnancy, age, etc. That is not a problem, but the authors should clearly state whether this was an ITT analysis. For the scabies clearance analysis, they should also provide a per protocol analysis. 

5. Were study team members who assessed adverse events shortly after treatment and presence/absence of scabies at 2 months and 12 months blinded with regard to treatment history? If not, what measures were taken to reduce possible bias in ascertainment of AEs or assessment of scabies? 

6. The authors suggest that more studies are needed to determine when to implement MDA vs. selective treatment for scabies. Given the high cost of the two step screening process and somewhat disappointing results in terms of new cases and persistent scabies at 2 and 12 months in the intervention area, I think that the Discussion should cite these results as a strong argument for MDA. It is clear that many cases of scabies are sub-clinical, and others might be missed in the first round of screening by community health workers. Some cases identified by the primary screen were not present for secondary screening or treatment, and that probably diminished the impact of the ivermectin intervention. It is likely that MDA would have provided superior results to selective treatment.

Reviewer #2: (No Response)

Reviewer #3: This is an important study, it’s the first published paper on a mass screening treatment approach for the control of scabies and as such, provides novel evidence. Its overall well designed within a cluster randomized approach and it is well written overall with appropriate discussions and conclusions

- The authors need to reference and set the study within the context of the recently published WHO framework on scabies control - this can be found here: https://www.who.int/publications/i/item/9789240008069

In addition to this there are some specific comments in relation to the introduction and typographical errors throughout the manuscript

- The authors should go through the manuscript again as there are many typographical errors such as double full stops in lines 39, 55 and 164, standardize space between numbers and brackets for example in line 251 1430(91.3%) vs 1113 (94.0%), Table 1 “No” vs “No.” Line 474 whould say “Identifying” rather than “Indetifying”

- The ending of the sentence in lines 101-103 sounds incomplete, suggest replacing “with excellent safety” to “safely” 

- The usage of reference 11 is incorrect, this is a letter in response to reference 10 and not a study about ivermectin MDA in Australia.

PLOS authors have the option to publish the peer review history of their article (what does this mean?). If published, this will include your full peer review and any attached files.

Reviewer #1: No

Reviewer #2: No

Reviewer #3: No
---

## [Decision Letter · Decision Letter 1]

24 Feb 2021

Dear Dr. Bang,

Thank you very much for submitting your manuscript "Control of scabies in a tribal community using mass screening and treatment with oral ivermectin -A cluster randomized controlled trial in Gadchiroli, India" for consideration at PLOS Neglected Tropical Diseases. As with all papers reviewed by the journal, your manuscript was reviewed by members of the editorial board and by several independent reviewers. The reviewers appreciated the attention to an important topic. Based on the reviews, we are likely to accept this manuscript for publication, providing that you modify the manuscript according to the review recommendations. 

Sincerely,

Arunasalam Pathmeswaran

Associate Editor

Dennis Bente

Deputy Editor

Reviewer's Responses to Questions

**Key Review Criteria Required for Acceptance?**

**Methods**

-Are the objectives of the study clearly articulated with a clear testable hypothesis stated?

-Is the study design appropriate to address the stated objectives?

-Is the population clearly described and appropriate for the hypothesis being tested?

-Is the sample size sufficient to ensure adequate power to address the hypothesis being tested?

-Were correct statistical analysis used to support conclusions?

-Are there concerns about ethical or regulatory requirements being met?

Reviewer #2: (No Response)

**Results**

-Does the analysis presented match the analysis plan?

-Are the results clearly and completely presented?

-Are the figures (Tables, Images) of sufficient quality for clarity?

Reviewer #2: (No Response)

**Conclusions**

-Are the conclusions supported by the data presented?

-Are the limitations of analysis clearly described?

-Do the authors discuss how these data can be helpful to advance our understanding of the topic under study?

-Is public health relevance addressed?

Reviewer #2: (No Response)

**Editorial and Data Presentation Modifications?**

Reviewer #2: (No Response)

**Summary and General Comments**

Reviewer #2: (No Response)

PLOS authors have the option to publish the peer review history of their article (what does this mean?). If published, this will include your full peer review and any attached files.

Reviewer #2: No
---

## [Editor Report · Decision Letter 2]

25 Mar 2021

Dear Dr. Bang,

We are pleased to inform you that your manuscript 'Control of scabies in a tribal community using mass screening and treatment with oral ivermectin -A cluster randomized controlled trial in Gadchiroli, India' has been provisionally accepted for publication in PLOS Neglected Tropical Diseases.

Best regards,

Arunasalam Pathmeswaran

Associate Editor

Dennis Bente

Deputy Editor

---

## [Editor Report · Acceptance letter]

8 Apr 2021

Dear Dr. Bang,

We are delighted to inform you that your manuscript, "Control of scabies in a tribal community using mass screening and treatment with oral ivermectin -A cluster randomized controlled trial in Gadchiroli, India," has been formally accepted for publication in PLOS Neglected Tropical Diseases.

Best regards,

Shaden Kamhawi

co-Editor-in-Chief

Paul Brindley

co-Editor-in-Chief
